# Regret Pre-training: Bridging Prior and Posterior Views for Enhanced Knowledge Grounding

**Mingkuan Zhao** [* 1]   **Xiayu Sun** [* 1]   **Wentao Hu** [1]   **Suquan Chen** [1]   **Jiaxuan Li** [1]   **Xiaoyan Zhu** [1]   **Xin Lai** [1]   **Jiayin Wang** [1]

## Abstract

Causal language models factorize sequence probabilities using only preceding context, leaving future information unexploited during training despite its availability in the training data. This paper introduces Regret Pre-training, a self-supervised framework grounded in the Learning Using Privileged Information (LUPI) paradigm. The framework employs a dual-view architecture in which a single model generates both a causal Student distribution and a future-conditioned Teacher distribution. The training objective augments standard language modeling with a regret loss that minimizes the KL divergence from teacher to student, transferring future-aware signals to the causal representations. We investigate two teacher configurations on the OLMoE-1B-7B architecture: LOCALREGRET, which extends attention by one future token, and GLOBALREGRET, which conditions on bidirectional context with the target position masked. Experiments on nine downstream tasks following 4 billion tokens of training demonstrate that both configurations consistently outperform the baseline. On average, GLOBALREGRET and LOCALREGRET achieve 33.9% and 32.2% accuracy respectively, surpassing the baseline's 30.2%. Most notably, GLOBAL-REGRET improves BoolQ performance by 18.1 percentage points (61.0% vs 42.9%). The framework introduces no additional parameters and requires only one extra inference-mode forward pass per training step. The source code for this paper is publicly available at https://github.com/RegretPretraining/Code2026.

---

[*]Equal contribution  [1]Contact email: xiayusun@stu.xjtu.edu.cn. The School of Computer Science and Technology, Xi'an Jiaotong University, China. Correspondence to: Jiayin Wang <wangjiayin@mail.xjtu.edu.cn>.

*Proceedings of the 43rd International Conference on Machine Learning*, Seoul, South Korea. PMLR 306, 2026. Copyright 2026 by the author(s).

## 1. Introduction

Large Language Models (LLMs) have achieved substantial progress in natural language understanding and generation tasks. The predominant training paradigm for these models is Causal Language Modeling (CLM), which factorizes the joint probability of a text sequence $X = (x_1, x_2, \ldots, x_T)$ as a product of conditional probabilities:

$$P(X) = \prod_{t=1}^{T} P(x_t | x_{<t}) \qquad (1)$$

Under this formulation, the probability of each token is conditioned exclusively on its preceding context (Vaswani et al., 2017; Touvron et al., 2023). This left-to-right factorization enables efficient autoregressive generation and has proven effective across a wide range of applications. However, an inherent asymmetry exists between training and inference: during training, the complete sequence $(x_1, x_2, \ldots, x_T)$ is fully observed, yet the model is trained to predict each token $x_t$ using only the partial observation $(x_1, \ldots, x_{t-1})$. The future context $(x_{t+1}, \ldots, x_T)$, though present in the training data, remains unexploited under the standard causal objective.

The exclusion of future context during training constitutes a form of information waste. Natural language exhibits statistical dependencies that extend beyond unidirectional causality. Coreference resolution, syntactic agreement, and semantic coherence often require bidirectional reasoning. When a human reader encounters an ambiguous token, they leverage both preceding and following context to resolve the ambiguity. The causal language model, by construction, cannot access this disambiguating information during training, despite its availability in the training corpus. This limitation becomes particularly pronounced in domains where long-range dependencies dominate local patterns. The question then arises whether future context can be incorporated into the training process without compromising the causal structure required for autoregressive generation.

This paper introduces *Regret Pre-training*, a framework that addresses this asymmetry by incorporating future context as privileged information during training. The term *regret*

originates from decision theory, where it quantifies the difference between the outcome of a decision made under uncertainty and the outcome that would have been achieved with full information (Savage, 1951). In language modeling, the causal model makes predictions under uncertainty about future tokens, while a model with access to future context can make more informed predictions. The *regret* of the causal model, in this sense, is the gap between its predictive distribution and the distribution that would be formed with knowledge of the future.

Our framework operationalizes this concept through a dual-view architecture. Let $P_S(x_t|x_{<t})$ denote the *Student* distribution, which maintains causal attention and corresponds to the standard autoregressive model. Let $P_T(x_t|x_{<t}, x_{>t})$ denote the *Teacher* distribution, which conditions on both past and future context. Both distributions are parameterized by the same model weights $\theta$, differing only in their attention masks. The training objective augments the standard negative log-likelihood with a *regret loss* that minimizes the KL divergence from teacher to student:

$$\mathcal{L}(\theta) = \underbrace{-\sum_t \log P_S(x_t|x_{<t})}_{\text{Causal Objective}} + \alpha \underbrace{\sum_t D_{\text{KL}}(P_T\|P_S)}_{\text{Regret Loss}} \quad (2)$$

This formulation instantiates Vapnik's Learning Using Privileged Information (LUPI) paradigm (Vapnik & Vashist, 2009), where the future context serves as privileged information available during training but not during inference. The regret loss transfers the teacher's future-informed predictions to the student, enabling the causal model to internalize patterns that would otherwise require future observation.

The information-theoretic foundation of this approach rests on the data processing inequality. For any position $t$, the mutual information between the input and target satisfies $I(x_{<t}, x_{>t}; x_t) \geq I(x_{<t}; x_t)$, with equality only when the future provides no additional information about the target given the past. In natural language, this inequality is typically strict: the future context $(x_{t+1}, \ldots, x_T)$ often disambiguates the target token $x_t$ in ways that the past context alone cannot. The regret loss encourages the student to extract features from the causal context that maximize correlation with the teacher's future-informed distribution, effectively distilling the information gain $I(x_{>t}; x_t|x_{<t})$ into the student's representations. (Kahneman, 2011)

A critical design dimension in this framework is the *scope* of future context available to the teacher. We investigate two configurations representing endpoints of this spectrum. The **LOCALREGRET** configuration restricts the teacher to observe exactly one future token, setting $\mathcal{C}_t = (x_1, \ldots, x_t, x_{t+1})$. The **GLOBALREGRET** configuration grants the teacher access to the complete sequence excluding the target position, setting $\mathcal{C}_t = (x_1, \ldots, x_t, x_{t+2}, \ldots, x_T)$.

These two configurations induce qualitatively different training dynamics: LOCALREGRET provides a local signal derived from immediate successor information, while GLOBALREGRET provides a global signal derived from long-range bidirectional context. The choice between these configurations determines the nature of the inductive bias transferred to the student model.

We conduct experiments on the OLMoE-1B-7B architecture (Muennighoff et al., 2024), a sparse Mixture-of-Experts model with 1 billion active parameters and 7 billion total parameters. Training is performed on the DCLM-Baseline corpus (Li et al., 2025) using 8 NVIDIA A100 GPUs. Evaluation spans nine downstream tasks in the zero-shot setting, covering reading comprehension, factual knowledge, commonsense reasoning, and natural language inference. The experimental results demonstrate that both LOCAL-REGRET and GLOBALREGRET consistently outperform the baseline causal language model across all evaluated benchmarks. The two configurations exhibit distinct performance profiles, with GLOBALREGRET achieving the largest improvements on reading comprehension tasks and LO-CALREGRET achieving the largest improvements on commonsense reasoning tasks. When averaged across all nine benchmarks, GLOBALREGRET improves accuracy by 3.7 percentage points over the baseline, and LOCALREGRET improves accuracy by 2.0 percentage points over the baseline. The largest single-task improvement is observed on BoolQ, where GLOBALREGRET achieves an accuracy of 61.0% compared to the baseline accuracy of 42.9%, representing an improvement of 18.1 percentage points.

The Regret Pre-training framework offers several desirable properties for large-scale language model training. The parameter sharing between teacher and student views eliminates the need for a separate teacher model, avoiding the memory overhead associated with traditional knowledge distillation methods. The teacher forward pass operates without gradient computation, adding only inference-time cost to each training step. The method requires no architectural modifications to the underlying transformer, with all changes confined to attention mask construction. The framework is compatible with standard training optimizations including sequence packing, mixed-precision arithmetic, and gradient accumulation. These properties enable direct integration with existing pre-training pipelines without requiring specialized infrastructure or significant computational overhead beyond the additional forward pass.

**Conflict of Interest Disclosure.** All authors of this work are affiliated with the School of Computer Science and Technology, Xi'an Jiaotong University. The models evaluated in this study, including the OLMoE-1B-7B architecture and the DCLM-Baseline corpus, are publicly released artifacts developed by third parties with which the authors have no fi-

nancial or employment relationship. The authors declare no financial conflicts of interest related to the research, methodology, or results presented in this paper.

## 2. Related Work

Research on distilling a bidirectional teacher into a unidirectional student was introduced by Chen et al. (2020) (Chen et al., 2020) within the context of Neural Machine Translation. They demonstrated that a BERT teacher could improve the generation quality of a causal decoder. However, this approach relies on maintaining a separate encoder model, which doubles the memory requirements. In the context of Large Language Models (LLMs) with hundreds of billions of parameters, maintaining a distinct teacher model is computationally infeasible. The work presented here extends this paradigm by integrating the teacher as a view of the same model weights. This method requires only an additional forward pass without backpropagation, making it efficient for training large-scale models.

Several architectures have been developed to incorporate future context directly into the training objective. Prophet-Net (Qi et al., 2020) modifies the self-attention mechanism to predict the next $N$ tokens simultaneously. Similarly, GLM (Du et al., 2022) utilizes a permutation objective to enable bidirectional attention over spans, while XLNet (Yang et al., 2019) employs permutation language modeling to access future contexts. Recent work has also explored sparse attention patterns to improve efficiency while maintaining performance (Zhao et al., 2026). Objectives like Fill-in-the-Middle (Bavarian et al., 2022) have successfully adapted causal models to utilize bidirectional context. Although these methods are effective, they often necessitate non-standard architectures or specific inference procedures. In contrast, our work focuses on objective modifications that leave the standard causal transformer architecture unchanged.

In the domain of diffusion and consistency models, CDLM (Strudel et al., 2022) employs a block-causal masking strategy to distill the iterative refinement capability of diffusion into autoregressive models. Furthermore, BiAlign (Qin et al., 2025) aligns causal models with bidirectional priors for In-Context Learning.

The theoretical foundation of our approach is derived from Vapnik's Learning Using Privileged Information (LUPI) framework (Vapnik & Vashist, 2009). The central concept of LUPI is defined by the inequality $I(X, X^*; Y) \geq I(X; Y)$, which states that the mutual information between the input and the target is increased given privileged information $X^*$. In this framework, $X^*$ represents the future text $x_{t+1:T}$. The objective of regret learning is to transfer the information gap defined by this inequality into the model parameters $\theta$.

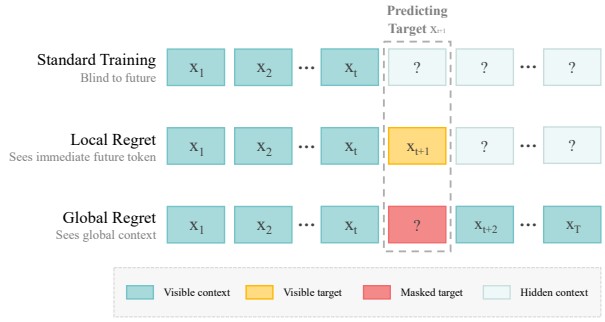

*Figure 1.* **Dual-View Regret Distillation.** The model processes each input under two attention configurations: a causal *Student View* and a future-aware *Teacher View*. The *Regret Loss* minimizes the KL divergence between the student distribution $P_S$ and the teacher distribution, where the teacher is conditioned on either local context ($P_{TL}$) or global context ($P_{TG}$).

Beyond modifications to the pre-training objective, a parallel line of research has sought to improve the efficiency and reliability of Mixture-of-Experts architectures through interventions on the expert pool itself. Mosaic Pruning (Hu et al., 2026b) addresses the substantial static memory overhead of sparse MoE models by introducing a hierarchical "cluster-then-select" framework that constructs a functionally comprehensive subset of experts, mitigating the catastrophic cross-domain degradation observed in single-corpus pruning approaches. Counterfactual Routing (Hu et al., 2026a) instead targets the inference-time behavior of MoE routers, demonstrating that hallucinations can be alleviated by awakening dormant experts whose specialized knowledge is otherwise bypassed under standard routing. These approaches share with our framework the broader goal of extracting greater capability from sparse architectures without inflating parameter counts, but operate along orthogonal axes: expert selection and routing rather than the training objective. Regret Pre-training is therefore complementary to these methods, and the structural sparsity they exploit could in principle be combined with objective-level distillation to further improve the trade-off between capability and compute.

## 3. Methodology

This section presents the Regret Pre-training framework. We first establish the mathematical formulation in Section 3.1, then describe the two teacher view configurations in Section 3.2, and finally detail the extension to packed sequences and Mixture-of-Experts architectures in Sections 3.3.

### 3.1. Mathematical Framework

Let $X = (x_1, x_2, \ldots, x_T)$ denote a sequence of tokens drawn from vocabulary $\mathcal{V}$. We consider a transformer model

$\pi_\theta$ parameterized by $\theta$ that defines conditional distributions over the vocabulary at each position.

### 3.1.1. DUAL-VIEW ARCHITECTURE

The proposed framework processes each training batch through two distinct computational paths using identical parameters $\theta$. The *Student View* applies a standard causal attention mask, producing the distribution:

$$P_S(x_{t+1}|x_{1:t};\theta) = \text{softmax}\left(f_\theta(x_{1:t})_t\right) \quad (3)$$

where $f_\theta(x_{1:t})_t \in \mathbb{R}^{|\mathcal{V}|}$ denotes the logit vector at position $t$ under causal attention. The *Teacher View* applies a modified attention mask that incorporates future context, producing:

$$P_T(x_{t+1}|\mathcal{C}_t;\theta) = \text{softmax}\left(g_\theta(\mathcal{C}_t)_t\right) \quad (4)$$

where $\mathcal{C}_t \supseteq x_{1:t}$ denotes the extended context available to the teacher, and $g_\theta(\mathcal{C}_t)_t$ denotes the corresponding logit vector. The teacher view operates in inference mode with all gradients detached.

### 3.1.2. COMPOSITE OBJECTIVE FUNCTION

The optimization objective combines the standard language modeling loss with a distillation term:

$$\mathcal{L}(\theta) = \mathcal{L}_{\text{NLL}}(\theta) + \alpha \cdot \mathcal{L}_{\text{Regret}}(\theta) \quad (5)$$

where $\alpha \geq 0$ is a hyperparameter. The negative log-likelihood component is defined as:

$$\mathcal{L}_{\text{NLL}}(\theta) = -\frac{1}{T}\sum_{t=1}^{T} \log P_S(x_{t+1}|x_{1:t};\theta) \quad (6)$$

This term ensures that the model retains standard autoregressive generation capability.

### 3.1.3. REGRET LOSS FORMULATION

The regret loss quantifies the divergence between the student and teacher distributions. We adopt the forward Kullback-Leibler divergence:

$$\mathcal{L}_{\text{Regret}}(\theta) = \frac{1}{T}\sum_{t=1}^{T} D_{\text{KL}}\left(\text{sg}[P_T(\cdot|\mathcal{C}_t)] \parallel P_S(\cdot|x_{1:t})\right) \quad (7)$$

where $\text{sg}[\cdot]$ denotes the stop-gradient operator. Expanding the KL divergence:

$$\begin{aligned} D_{\text{KL}}(P_T \parallel P_S) &= \sum_{v \in \mathcal{V}} P_T(v) \log \frac{P_T(v)}{P_S(v)} \\ &= H(P_T, P_S) - H(P_T) \end{aligned} \quad (8)$$

where $H(P_T, P_S)$ is the cross-entropy and $H(P_T)$ is the entropy of the teacher distribution. Since the teacher parameters are detached from the computational graph, $H(P_T)$ is

constant with respect to $\theta$, and minimizing Equation 7 is equivalent to minimizing:

$$\mathcal{L}_{\text{Regret}}(\theta) \equiv \frac{1}{T}\sum_{t=1}^{T} H\left(\text{sg}[P_T(\cdot|\mathcal{C}_t)], P_S(\cdot|x_{1:t})\right) \quad (9)$$

### 3.1.4. CHOICE OF DIVERGENCE DIRECTION

The forward KL divergence $D_{\text{KL}}(P_T \parallel P_S)$ and reverse KL divergence $D_{\text{KL}}(P_S \parallel P_T)$ exhibit distinct optimization behaviors. The gradient of the forward KL with respect to the student logits $z_S$ takes the form:

$$\frac{\partial D_{\text{KL}}(P_T \parallel P_S)}{\partial z_S} = P_S - P_T \quad (10)$$

This gradient is bounded and well-defined across the entire support of $P_T$. In contrast, the reverse KL gradient:

$$\frac{\partial D_{\text{KL}}(P_S \parallel P_T)}{\partial z_S} = P_S\left(\log P_S - \log P_T + 1\right) \quad (11)$$

contains the term $\log P_T$, which becomes unbounded as $P_T(v) \to 0$ for any $v \in \mathcal{V}$. The forward KL formulation therefore provides stable gradients throughout training.

### 3.1.5. CONNECTION TO LEARNING USING PRIVILEGED INFORMATION

The proposed framework instantiates the Learning Using Privileged Information (LUPI) paradigm (Vapnik & Vashist, 2009). In LUPI, a teacher has access to privileged information $X^*$ unavailable at inference time. The data processing inequality establishes:

$$I(X, X^*; Y) \geq I(X; Y) \quad (12)$$

where $I(\cdot; \cdot)$ denotes mutual information. In our formulation, $X = x_{1:t}$ represents the causal context, $X^* = x_{t+1:T}$ represents the future context, and $Y = x_{t+1}$ is the prediction target. The teacher distribution $P_T(Y|X, X^*)$ conditions on both past and future, while the student $P_S(Y|X)$ conditions only on the past. The regret loss transfers information from the privileged view to the student's representations.

### 3.2. Teacher View Configurations

The teacher's attention mask $M^{\text{teacher}} \in \mathbb{R}^{T \times T}$ determines the context $\mathcal{C}_t$ at each position. We define two configurations representing distinct points in the design space of future context scope.

### 3.2.1. LOCAL CONFIGURATION (LOCALREGRET)

The local configuration extends the causal context by exactly one position. For a query at position $i$, the attention mask

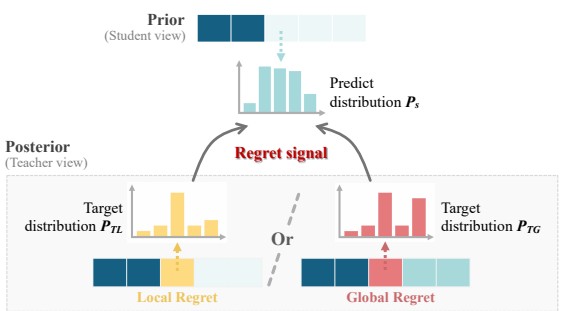

*Figure 2.* **Teacher View Configurations.** Three attention patterns are illustrated. **Standard (Causal)** conditions on $x_{1:t}$. **LOCAL-REGRET** conditions on $x_{1:t+1}$. **GLOBALREGRET** conditions on $x_{1:t} \cup x_{t+2:T}$.

permits attending to all positions $j \leq i + 1$:

$$M_{i,j}^{\text{Local}} = \begin{cases} 0 & \text{if } j \leq i + 1 \\ -\infty & \text{otherwise} \end{cases} \quad (13)$$

The resulting teacher context is $\mathcal{C}_t = x_{1:t+1}$, and the teacher distribution becomes:

$$P_{TL}(x_{t+1}|x_{1:t+1}; \theta) = \text{softmax}\left(g_\theta(x_{1:t+1})_t\right) \quad (14)$$

Let $z_t \in \mathbb{R}^{|\mathcal{V}|}$ denote the student logits at position $t$. Under the local configuration, the teacher logits $\tilde{z}_t$ incorporate attention to position $t + 1$. The teacher distribution can be expressed as:

$$P_{TL}(v|x_{1:t+1}) = \frac{\exp(\tilde{z}_t^{(v)})}{\sum_{v' \in \mathcal{V}} \exp(\tilde{z}_t^{(v')})} \quad (15)$$

The regret loss for this configuration is:

$$\mathcal{L}_{\text{Regret}}^{\text{Local}} = \frac{1}{T} \sum_{t=1}^{T} \sum_{v \in \mathcal{V}} P_{TL}(v|x_{1:t+1}) \log \frac{P_{TL}(v|x_{1:t+1})}{P_S(v|x_{1:t})} \quad (16)$$

### 3.2.2. GLOBAL CONFIGURATION (GLOBALREGRET)

The global configuration grants bidirectional attention to the full sequence while masking the target position. For a query at position $i$ predicting position $i + 1$, the mask is:

$$M_{i,j}^{\text{Global}} = \begin{cases} -\infty & \text{if } j = i + 1 \\ 0 & \text{otherwise} \end{cases} \quad (17)$$

The teacher context becomes $\mathcal{C}_t = x_{1:t} \cup x_{t+2:T}$, yielding the distribution:

$$P_{TG}(x_{t+1}|x_{1:t}, x_{t+2:T}; \theta) = \text{softmax}\left(g_\theta(x_{1:t}, x_{t+2:T})_t\right) \quad (18)$$

The information available to the teacher under the global configuration satisfies $I(x_{1:t}, x_{t+2:T}; x_{t+1}) \geq I(x_{1:t}; x_{t+1})$ by the data processing inequality.

### 3.2.3. ATTENTION COMPLEXITY ANALYSIS

Both configurations maintain the same asymptotic complexity as standard causal attention. Let $T$ denote sequence length and $d$ denote model dimension. For LOCALREGRET, the attention pattern remains lower-triangular with one additional diagonal, requiring $O(T^2 d)$ operations. For GLOBALREGRET, the attention pattern is dense with $T$ masked positions (one per row), requiring $O(T^2 d)$ operations. The total training cost per batch increases by a factor of at most $2\times$ due to the additional teacher forward pass, with no increase in memory footprint for model parameters.

### 3.3. Extension to Packed Sequences

Training pipelines for large language models concatenate multiple documents into fixed-length sequences to maximize hardware utilization. Let a packed sequence of length $L$ contain documents with lengths $D = (d_1, d_2, \ldots, d_K)$ where $\sum_{k=1}^{K} d_k = L$. The attention mask must enforce document boundaries to prevent cross-document attention.

### 3.3.1. MASK CONSTRUCTION FOR GLOBALREGRET

For the global configuration with packed sequences, the attention mask combines target-position masking with document isolation. Define the document assignment function $\phi : \{1, \ldots, L\} \rightarrow \{1, \ldots, K\}$ mapping each position to its document index:

$$\phi(i) = k \quad \text{where} \quad \sum_{j=1}^{k-1} d_j < i \leq \sum_{j=1}^{k} d_j \quad (19)$$

The composite mask is then:

$$M_{i,j}^{\text{Packed}} = \begin{cases} -\infty & \text{if } j = i + 1 \text{ and } \phi(i) = \phi(j) \\ -\infty & \text{if } \phi(i) \neq \phi(j) \\ 0 & \text{otherwise} \end{cases} \quad (20)$$

The first condition implements target-position masking within documents, and the second condition enforces document isolation.

Algorithm 1 presents an efficient vectorized implementation. The algorithm constructs the mask in $O(L^2)$ time using broadcasting operations, enabling integration with standard training frameworks.

### 3.3.2. MASK CONSTRUCTION FOR LOCALREGRET

For the local configuration, the mask extends causal attention by one position while respecting document boundaries:

$$M_{i,j}^{\text{Packed-Local}} = \begin{cases} -\infty & \text{if } j > i + 1 \\ -\infty & \text{if } \phi(i) \neq \phi(j) \\ 0 & \text{otherwise} \end{cases} \quad (21)$$

**Algorithm 1** Attention Mask Construction for GLOBALRE-
GRET with Packed Sequences

---

1: **Input:** Sequence length $L$, document lengths $D = (d_1, \ldots, d_K)$
2: **Output:** Attention bias matrix $M \in \mathbb{R}^{1 \times 1 \times L \times L}$
3: Initialize $M \leftarrow \mathbf{0}^{L \times L}$
4: Compute position indices $\mathbf{p} \leftarrow (0, 1, \ldots, L-1)$
5: {Construct target-position mask}
6: $T_{i,j} \leftarrow \mathbf{1}[j = i+1]$ for all $i, j \in \{0, \ldots, L-1\}$
7: {Construct document assignment vector}
8: $\phi \leftarrow \text{repeat}((0, 1, \ldots, K-1), (d_1, \ldots, d_K))$
9: {Construct document isolation mask}
10: $B_{i,j} \leftarrow \mathbf{1}[\phi_i \neq \phi_j]$ for all $i, j \in \{0, \ldots, L-1\}$
11: {Combine masks}
12: $M_{i,j} \leftarrow -\infty$ where $T_{i,j} = 1$ or $B_{i,j} = 1$
13: **return** $M$ reshaped to $(1, 1, L, L)$

---

This ensures that the lookahead attention does not cross document boundaries within the packed sequence.

## 4. Experiments

We empirically validate the proposed Regret Pre-training framework using the OLMoE architecture. The experimental design focuses on quantifying the downstream zero-shot performance benefits of integrating future-conditioned regret signals into the pre-training objective. We rigorously control for model capacity, training data, and computational constraints to ensure a fair comparison between the baseline causal approach and our dual-view distillation methods.

### 4.1. Experimental Setup

**Model Architecture and Data** We utilize the OLMoE-1B-7B architecture, a sparse Mixture-of-Experts (MoE) model comprising 1 billion active parameters and approximately 7 billion total parameters. The architecture employs a router mechanism to select experts for each token, allowing for high total capacity with reduced inference costs. All configurations are pre-trained from scratch on a subset of the DCLM-Baseline corpus. The training budget is fixed at 4 billion tokens for all models to simulate a compute-constrained pre-training scenario. To maximize hardware utilization, we employ sequence packing, where multiple documents are concatenated to fill a fixed sequence length of 2,048 tokens. The model architecture consists of 16 transformer layers with a hidden dimension of 2,048 and 16 attention heads. Each MoE layer contains 64 experts with a top-2 routing strategy.

**Training Implementation** Experiments are conducted on a cluster of 8 NVIDIA A100 (80GB) GPUs. We implement the training loop using PyTorch with mixed-precision

(bfloat16) arithmetic. The optimization process involves two distinct forward passes per training step. The first pass computes the standard causal language modeling loss and the auxiliary load-balancing loss required for the MoE router. The second pass computes the teacher distribution under the specified future-aware attention mask. Crucially, the teacher forward pass is executed in inference mode with gradient computation disabled to minimize memory overhead. We employ the AdamW optimizer with a learning rate of $3 \times 10^{-4}$, weight decay of 0.1, and gradient clipping with a maximum norm of 1.0. The learning rate follows a cosine decay schedule with 2,000 warmup steps. Training runs for 50,000 steps, corresponding to approximately 4 billion tokens given the sequence length of 2,048 and batch size configuration.

**MoE Routing Isolation** A critical implementation detail in Regret Pre-training with MoE architectures is the handling of the auxiliary load-balancing loss. The teacher view, having access to future context, may generate routing decisions that differ significantly from those of a causal model. Including the teacher's routing statistics in the auxiliary loss calculation would optimize the router for a data distribution unavailable during inference. We therefore explicitly clear the load-balancing statistics prior to the teacher forward pass and compute the auxiliary loss exclusively based on the student's routing decisions:

$$\mathcal{L}_{\text{aux}}(\theta) = \sum_{e=1}^{E} f_e^{(S)} \cdot p_e^{(S)} \tag{22}$$

where $f_e^{(S)}$ denotes the fraction of tokens routed to expert $e$ and $p_e^{(S)}$ denotes the average routing probability for expert $e$, both computed from the student forward pass. This ensures that the expert selection mechanism is optimized solely for causal inference. The auxiliary loss is weighted by a coefficient of 0.01 and added to the total training objective.

**Attention Masking and Document Isolation** We strictly enforce document boundaries within packed sequences to prevent data leakage. For the GLOBALREGRET configuration, we construct a composite attention mask that serves two functions. First, it permits bidirectional attention while masking the specific target position $t+1$ for a query at $t$. Second, it applies a block-diagonal mask based on document identifiers. Let $D(i)$ denote the document index of token $i$. The attention weight $A_{ij}$ is masked if $D(i) \neq D(j)$, regardless of the relative positions of $i$ and $j$. This document isolation ensures that the teacher's privileged information is strictly limited to the future context of the current document. We utilize a masking value of $-10^4$ rather than negative infinity to ensure numerical stability within the bfloat16 dynamic range while effectively zeroing out attention probabilities after the softmax operation. The packed sequences

contain between 3 and 8 documents per batch depending on individual document lengths, with padding applied only at sequence boundaries.

## 4.2. Evaluation Protocol

We evaluate the pre-trained models on a diverse suite of nine downstream benchmarks in a zero-shot setting. The evaluation set covers reading comprehension (BoolQ), scientific reasoning (ARC-Challenge), commonsense reasoning (PIQA, HellaSwag, WinoGrande, CommonsenseQA), and multi-task language understanding (MMLU). For the MMLU benchmark, we report results across four aggregated categories: Humanities, Other, Social Sciences, and STEM. We employ a likelihood-based evaluation metric where the model computes the conditional probability of each answer option given the context, selecting the option with the highest likelihood. This protocol assesses the intrinsic knowledge grounding of the model without introducing variance from generation sampling strategies or few-shot prompt engineering. All evaluations are performed using the EleutherAI Language Model Evaluation Harness framework with default settings. For multiple-choice tasks, we compute the log-likelihood of each candidate continuation and normalize by the number of tokens in each option to account for length bias. The final prediction corresponds to the option with the highest normalized log-likelihood.

## 5. Results

Table 1 summarizes the zero-shot performance of the baseline causal model compared to the LOCALREGRET and GLOBALREGRET configurations. The baseline model is trained using the standard autoregressive objective for 4 billion tokens. The regret-based models utilize the same training data and budget, differing only in the inclusion of the auxiliary regret loss with $\alpha = 1.0$.

Both regret-based configurations outperform the baseline across all seven individual tasks, demonstrating the consistent effectiveness of incorporating future-conditioned signals during pre-training. The average accuracy across all benchmarks increases from 30.2% for the baseline to 32.2% for LOCALREGRET and 33.9% for GLOBALREGRET. These improvements are achieved without any increase in model parameters or architectural modifications, validating the efficiency of the dual-view distillation framework.

The most substantial improvement is observed on BoolQ, where GLOBALREGRET achieves 61.0% accuracy compared to 42.9% for the baseline, representing an absolute gain of 18.1 percentage points. BoolQ requires models to answer binary yes/no questions based on passage content, a task that fundamentally depends on synthesizing information across the entire context to resolve the query. The

bidirectional attention available to the global teacher enables it to capture long-range dependencies that span the question and answer passages.

On tasks requiring global context integration, GLOBALREGRET consistently achieves the highest accuracy. For ARC-Challenge, which tests scientific reasoning and factual knowledge, GLOBALREGRET achieves 25.8% compared to 23.4% for LOCALREGRET and 21.7% for the baseline. On WinoGrande, which evaluates commonsense reasoning through pronoun disambiguation, GLOBALREGRET achieves 50.8%, LOCALREGRET achieves 50.6%, and the baseline achieves 49.6%. The superior performance of GLOBALREGRET on these benchmarks indicates that tasks requiring long-range reasoning and coreference resolution benefit substantially from bidirectional context signals.

Conversely, LOCALREGRET achieves the highest accuracy on tasks emphasizing local coherence and commonsense plausibility. On PIQA, which tests physical commonsense reasoning, LOCALREGRET achieves 58.9% compared to 57.2% for GLOBALREGRET and 56.4% for the baseline. On HellaSwag, which requires selecting the most likely continuation of a scenario, LOCALREGRET achieves 27.5% compared to 25.1% for GLOBALREGRET and 24.3% for the baseline. On CommonsenseQA, LOCALREGRET achieves 26.5% compared to 23.4% for GLOBALREGRET and 22.1% for the baseline, representing a relative improvement of 19.9% over the baseline. On MMLU, which aggregates performance across 57 tasks spanning multiple domains, LOCALREGRET achieves 25.8% compared to 25.0% for GLOBALREGRET and 24.5% for the baseline. These results demonstrate that local future context provides stronger training signals for tasks that depend on immediate transition dynamics and next-token prediction.

This dichotomy reflects the fundamental difference in the training signals provided by the two configurations. The global teacher distills long-range dependencies and bidirectional context into the student's representations, while the local teacher emphasizes immediate transition dynamics and next-step prediction. The choice of teacher configuration thus determines the inductive bias transferred to the causal model, with global configurations favoring tasks that require global reasoning and local configurations favoring tasks that require local plausibility judgments. The performance differential across task categories suggests that the optimal teacher configuration depends on the target application domain and the nature of reasoning required.

The computational overhead of regret pre-training is strictly limited to the additional forward pass required to compute the teacher distribution. Since the teacher operates in inference mode with all gradients detached, the memory footprint remains identical to standard training. The wall-clock time per training step increases by a factor of approximately 1.8

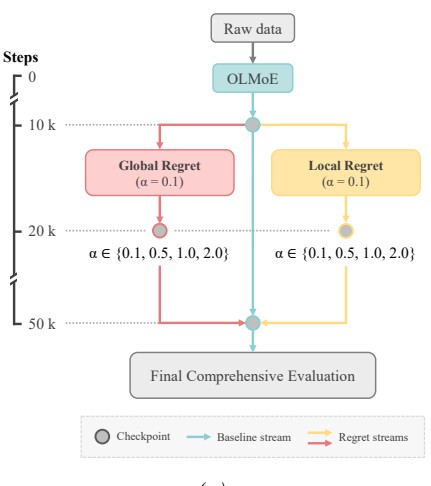

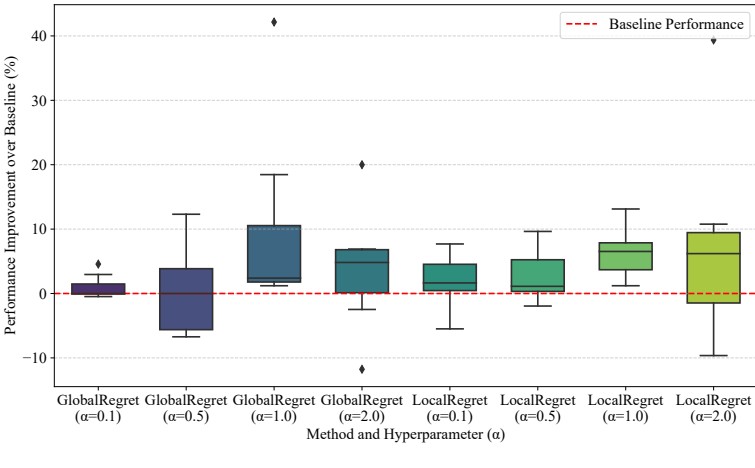

*(a)*            *(b)*

*Figure 3.* **Training Curriculum and Hyperparameter Sensitivity.** (a) Schematic of the branched training schedule designed to decouple stability from alignment gains. (b) Distribution of downstream task improvements across varying regret weights $\alpha$.

*Table 1.* Zero-shot accuracy comparison on downstream tasks after 4 billion tokens of pre-training. LOCALREGRET and GLOBALREGRET denote the proposed regret pre-training configurations. The Baseline represents standard causal language modeling. The highest accuracy for each task is highlighted in bold.

| METHOD | BOOLQ | ARC-C | WINOGRANDE | PIQA | HELLASWAG | CSQA | MMLU (AVG) | AVERAGE |
|---|---|---|---|---|---|---|---|---|
| BASELINE | 42.9 | 21.7 | 49.6 | 56.4 | 24.3 | 22.1 | 24.5 | 30.2 |
| LOCALREGRET | 48.5 | 23.4 | 50.6 | **58.9** | **27.5** | **26.5** | **25.8** | 32.2 |
| GLOBALREGRET | **61.0** | **25.8** | **50.8** | 57.2 | 25.1 | 23.4 | 25.0 | **33.9** |

to 2.0 depending on hardware and implementation details. This overhead is substantially lower than traditional knowledge distillation approaches that maintain separate teacher and student models, which typically require at least twice the memory and computational resources. Profiling measurements indicate that the teacher forward pass accounts for 45% of the total step time, with the remaining overhead attributed to loss computation and data preparation. The memory consumption per GPU remains at approximately 72GB throughout training, leaving sufficient headroom for the 80GB capacity of the A100 GPUs.

To justify the selection of the hyperparameter $\alpha$, we empirically investigate the impact of the regret loss magnitude on downstream zero-shot performance. As shown in Figure 3a, training begins with a standard causal warmup. At step 10k, we branch into Local and Global streams with unified ($\alpha = 0.1$). At step 20k, we perform a sensitivity sweep, branching into varying regret strengths $\alpha \in \{0.1, 0.5, 1.0, 2.0\}$ to analyze the performance and stability boundaries. Each branch is trained for an additional 30,000 steps to reach the final checkpoint at step 50,000.

Figure 3b illustrates the distribution of accuracy improvements relative to the baseline across the benchmarks. The data reveals a non-linear relationship between the distilla-

tion strength and model efficacy. For the GLOBALREGRET configuration, the setting $\alpha = 1.0$ yields the highest median improvement while maintaining a favorable interquartile range. In contrast, lower coefficients (e.g., $\alpha = 0.1$) result in negligible deviations from the baseline, suggesting insufficient information transfer from the teacher distribution. Conversely, increasing coefficient to $\alpha = 2.0$ leads to performance degradation in the lower quartile and the emergence of negative outliers, indicating that excessive regularization may interfere with the optimization of the primary causal objective.(Wei et al., 2022; Power et al., 2022) The LOCALREGRET configuration demonstrates higher resilience to hyperparameter variations but similarly exhibits optimal convergence properties at $\alpha = 1.0$. Based on these observations, $\alpha = 1.0$ is adopted for all subsequent comparative experiments to maximize expected performance gain while mitigating instability.

## 6. Conclusion

This paper presents Regret Pre-training, a self-supervised framework that augments causal language modeling with future-aware distillation signals. The proposed method introduces a dual-view architecture in which a single transformer processes each input under both causal and future-

conditioned attention masks, enabling knowledge transfer from the privileged teacher view to the student view without additional model parameters.

We investigate two teacher view configurations that define distinct scopes of future context. GLOBALREGRET, which conditions the teacher on bidirectional context excluding the target position, achieves substantial improvements on reading comprehension and factual knowledge tasks. On BoolQ, GLOBALREGRET improves accuracy by 18.1 percentage points over the baseline, representing the largest gain observed in our experiments. LOCALREGRET, which extends the teacher's context by exactly one future token, achieves the highest performance on physical commonsense and commonsense reasoning tasks, improving PIQA, HellaSwag, and CSQA accuracy over the baseline.

The empirical results demonstrate that the scope of future context available to the teacher view yields distinct downstream performance profiles. Both configurations improve over standard causal language modeling on the majority of evaluated tasks, with each configuration achieving the highest accuracy on a complementary subset of the evaluation suite. The framework requires only an additional forward pass per training step without backpropagation, maintaining computational efficiency while enabling effective transfer of future-aware signals to the causal model.

The proposed approach establishes a new direction for self-supervised language model training that leverages the duality between causal and non-causal views of the same model weights. Future work may explore intermediate points in the spectrum of future context scope, alternative divergence measures for the regret loss, and scaling behavior on larger model architectures and training budgets.

## Impact Statement

This paper presents work whose goal is to advance the field of Machine Learning. There are many potential societal consequences of our work, none which we feel must be specifically highlighted here.

## Author Contributions

Mingkuan Zhao conceived and designed the core idea of the Regret Pre-training framework. Xiayu Sun implemented and carried out all of the experiments, and produced all of the figures in this paper. Wentao Hu, Suquan Chen, and Jiaxuan Li contributed to a portion of the manuscript text. Xiaoyan Zhu and Xin Lai assisted with manuscript polishing and revision. Jiayin Wang is the corresponding author.

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
