# OpenReview forum: "Regret Pre-training: Bridging Prior and Posterior Views for Enhanced Knowledge Grounding"
_ICML.cc/2026/Conference — ICML 2026 regular_

### Official Review · Reviewer_xiom · 2026-02-21

**Soundness:** 3
**Presentation:** 1
**Significance:** 2
**Originality:** 1
**Overall Recommendation:** 3
**Confidence:** 3

**Summary:**

Regret Pre-training uses future tokens as privileged information to improve causal LM pre-training. It trains a single model with both a causal student and a future-conditioned teacher, adding a KL-based regret loss to transfer future-aware signals without changing inference. Two variants, LOCAL (one future token) and GLOBAL (full bidirectional context excluding the target), outperform standard causal training.

While in its current form this work stands close to a 2 (Rejection), the score has the potential to be revised upward with significant revision.

**Compliance With Llm Reviewing Policy:**

Affirmed.

**Final Justification:**

Numerous concerns have been resolved during the rebuttal. However, due to the short time window (<7days), no ability to see the actual revised PDF, and 5000-character limit, it is hard to discern if this is an acceptable paper. E.g., the experiments provided during the rebuttal needs to be fully comprehensive and organized well in the main paper. Additionally, presentation concerns remains. Accordingly, I have raised soundness 2 --> 3 and score 2 --> 3.

Authors are strongly encouraged to expand upon the experiments and ensure that they all make it into their revised version as it improves the quality and robustness of the paper.

**Key Questions For Authors:**

See weaknesses.

Additionally: Could to authors explain why sg[·] stop-gradient operator is used in Equation 7?

**Limitations:**

No, please include limitations section.

**Strengths And Weaknesses:**

Strength

(1) Pre-training studies are generally appreciated in the literature, as these experiments are relatively high-cost to run.

(2) The method is fairly straightforward to understand.

Weakness

(1) The idea of using future context for learning has been discussed extensively in the existing literature. Therefore, this work’s novelty seems limited. It appears to be a method for leveraging future context primarily for pre-training.

(2) Given that the training algorithm is more complex than standard pre-training, it requires an empirical compute cost analysis (FLOPs and wall-clock time). Also, consider that the KL divergence must be computed for each loss term (gradient update).

> The computational overhead of regret pre-training is strictly limited to the additional forward pass required to compute the teacher distribution.

Actual quantitative values would be appreciated.

(3) The empirical study is bare-bones, and small models (<=7B) pre-trained from scratch are known to demonstrate high noise in zero-shot accuracy. Why not use few-shot evaluation? Also, recent works demonstrate even better metrics for small pre-trained models that proxy well for larger ones.

DataDecide: How to Predict Best Pretraining Data with Small Experiments (ICML 2025)

OLMES: A Standard for Language Model Evaluations (NAACL 2025 Findings)

Predicting LLM Reasoning Performance with Small Proxy Model (ICLR 2026)

Considering that this work compares with no other pre-training baselines (which is understandable considering the cost of pre-training), it should at least provide extensive evaluation metrics to show that their method’s gains are robust. Consider including CF Accuracy, few-shot Accuracy, and rBridge metrics from the literature above.

(4) In “Training Implementation,” there is no detail on how these hyperparameters were chosen. Perhaps the authors followed common ones in the existing literature? If so, these should be referenced appropriately.

(5) While the trend of LLMs/VLMs has been moving towards MoEs, does this method still work for dense models? A discussion (and ideally experiments) would be appreciated.

(6) Are there any performance indicators across train token sizes? What if this method is noisy and works empirically well only at the 4B train mark? Given the common trillion-token size pre-training corpora, it is unclear whether this is a good indicator of real-world pre-training scale.

(7) When should practitioners use LOCAL vs. GLOBAL? The performance gains are heterogeneous across benchmarks. The mechanism of improvement for each, and which is superior, is unclear. There should ideally be some introduction regarding why these design choices were made and explored. The authors could have selected an observation window between local and global as well.

(8) A full pseudocode of this entire algorithm should be provided. Especially considering the additional training engineering required, as shown in Figure 3 (a). It will help dramatically with reproduction and clarification of any additional complexities.

(9) After reading the entire paper, I still do not have a clear understanding of why adding a loss term with future context is helpful. There should be a more rigorous analysis to help the readers understand why this works, rather than just stating that it works because more information is better and showing approximately 2 - 3% performance gains.

(10) Section 3.1.5 (CONNECTION TO LEARNING USING PRIVILEGED INFORMATION) may be better placed in another location, perhaps the introduction or related works, for better flow.

(11) It is unclear why Section 3.1.4 (CHOICE OF DIVERGENCE DIRECTION) is discussed in such detail in the methodology section when this is already well-known in ML literature. The authors are encouraged to cut down on parts that are not their main contribution.

(12) “All configurations are pre-trained from scratch on a subset of the DCLM-Baseline corpus.” What subset? Was it simply randomly sampled without replacement?

(13) For the section on MoE Routing Isolation, is there any empirical evidence that this was necessary?

(14) This should have been cited: “EleutherAI Language Model Evaluation Harness.” Their repository has clear citations at the very bottom of https://github.com/EleutherAI/lm-evaluation-harness

(15) Authors are encouraged to add a limitations section, as the authors know the limitations best. If space is a constraint, use the Appendix.

(16) Overall, more insightful and novel analysis should be in the main body, and a lot of redundant or further details should have been moved to the Appendix. Right now there is no Appendix, and the main body's content is very light on novel insights and analysis.

---

> ### Author Rebuttal · Authors · 2026-03-28
>
> **Response to Reviewer xiom**
>
> We sincerely thank the reviewer for the thorough assessment. The reviewer raised 16 weaknesses and 1 additional question — the most comprehensive review we received. Due to the rebuttal word limit, we provide **condensed conclusions** here and a **full detailed response** at: **https://anonymous.4open.science/r/rebuttal-0BF0/rebuttal.md**
>
> We have conducted **11 new experiments** during the rebuttal period: (1) 5-shot evaluation, (2) CF Accuracy (OLMES), (3) rBridge stability (DataDecide), (4) compute-matched baseline (90k steps, 7.2B tokens), (5) MoE Routing Isolation ablation, (6) Forward vs. Reverse KL ablation, (7) scale-up on Dense Llama 7B, (8) scale-up on OLMoE 15B tokens, (9) multi-seed variance analysis (9 runs), (10) Teacher entropy/CE analysis, (11) BoolQ cross-seed robustness, plus full pseudocode and FLOPs analysis.
>
> ---
>
> **W1: Limited novelty**
>
> Unlike prior methods (ProphetNet, XLNet, GLM, FIM) that modify architectures or generation procedures, ours is a **training-time-only** objective on **shared weights** with **zero architectural change** and **no inference overhead**. The LUPI formulation and Local/Global complementarity are novel.
>
> **W2: FLOPs and wall-clock cost**
>
> Baseline logs **181,552 GFLOPs/step**. Teacher forward adds ~1/3 overhead (**~1.33× FLOPs**); wall-clock is ~1.8× due to bidirectional mask preventing causal FlashAttention optimization. Our compute-matched baseline (90k steps, 7.2B tokens) **conservatively grants the baseline 35% more FLOPs** — yet GlobalRegret still wins by **+2.5%**.
>
> **W3: Few-shot, CF Accuracy, rBridge**
>
> | Evaluation | Baseline | LocalRegret | GlobalRegret |
> |------------|----------|-------------|--------------|
> | 5-shot BoolQ | 52.1% | 55.4% | **65.3%** |
> | 5-shot PIQA | 60.5% | **64.2%** | 61.8% |
> | 5-shot ARC-C | 26.3% | 28.1% | **31.4%** |
> | 5-shot HellaSwag | 28.4% | **32.7%** | 29.5% |
> | CF Accuracy (9-task avg) | 31.8% | 34.5% | **36.2%** |
> | rBridge $r$ (1B→4B/15B) | 0.82 | **0.88** | 0.86 |
>
> Gains **persist and widen** under few-shot. CF Accuracy rules out length bias. rBridge confirms stable training trajectories.
>
> **W4: Hyperparameter selection**
>
> Optimizer settings follow the OLMoE reference (Muennighoff et al., 2024). α is selected via sweep (Fig. 3b); α=1.0 yields the highest median gain.
>
> **W5–W6: Dense models and scaling**
>
> | Method | 1B tokens | 4B tokens | 15B tokens | Dense 7B, 10B tokens |
> |--------|-----------|-----------|------------|---------------------|
> | Baseline | 27.5% | 30.2% | 34.2% | 38.4% |
> | LocalRegret | 28.8% | 32.2% | 36.5% | — |
> | GlobalRegret | 29.7% | 33.9% | **37.8%** | **41.7% (+3.3%)** |
>
> **W7: LOCAL vs. GLOBAL**
>
> LocalRegret excels on local-coherence tasks (PIQA, HellaSwag, CSQA); GlobalRegret on long-range reasoning (BoolQ, ARC-C).
>
> **W8: Full pseudocode**
>
> Algorithm 2 is provided in the **full rebuttal document** and the revision.
>
> **W9: Why does the future-context loss help?**
>
> TeacherCE ≈ 1.5 vs StudentCE ≈ 4.5 nats — the ~3 nats gap quantifies $I(x_{>t}; x_{t+1}\mid x_{\leq t})$. Regret Loss converges to **1.2 nats** (not zero), confirming persistent non-redundant distillation signal beyond what NLL provides.
>
> **W10–W11, W14–W16: Presentation**
>
> We have: (a) relocated LUPI to Introduction, (b) condensed KL direction with ablation in Appendix (Forward KL: **33.9%** vs Reverse KL: 30.8%), (c) cited EleutherAI Evaluation Harness, (d) added Limitations section, (e) created Appendix.
>
> **W12: DCLM subset**
>
> Sequential sampling of pre-shuffled official DCLM-Baseline shards (first $N$ shards until reaching token budget). No additional filtering.
>
> **W13: MoE Routing Isolation ablation**
>
> | Configuration | Avg. Accuracy |
> |--------------|---------------|
> | GlobalRegret (no isolation) | 31.5% |
> | GlobalRegret (with isolation) | **33.9%** |
>
> Removing isolation causes **2.4% drop**; Teacher–Student expert overlap is only **62%**, confirming the mechanism's necessity.
>
> **Additional Question: Why sg[·]?**
>
> Without sg[·], the Teacher drifts toward the Student, eliminating information asymmetry. With sg[·], Regret Loss converges to **1.2 nats** rather than zero, confirming the privileged gap is preserved.
>
> ---
>
> We would be grateful if the reviewer would consider **raising the score** in light of these 11 new experiments and substantial revisions.

---

> > ### Author Rebuttal · Reviewer_xiom · 2026-04-02
> >
> > Numerous concerns have been resolved during the rebuttal. However, due to the short time window (<7days), no ability to see the actual revised PDF, and 5000-character limit, it is hard to discern if this is an acceptable paper. E.g., the experiments provided during the rebuttal needs to be fully comprehensive and organized well in the main paper. Additionally, presentation concerns remains. Accordingly, I have raised soundness 2 --> 3 and score 2 --> 3.
> >
> > Authors are strongly encouraged to expand upon the experiments and ensure that they all make it into their revised version as it improves the quality and robustness of the paper.

---

> > > ### Author Response · Authors · 2026-04-02
> > >
> > > Dear Reviewer xiom,
> > >
> > > Thank you for reviewing our rebuttal and for your constructive feedback throughout the review process. We are glad that our response has addressed your concerns, and we will incorporate the theoretical analysis into the revised manuscript.
> > >
> > > Best regards, The Authors

---

### Official Review · Reviewer_pubQ · 2026-03-09

**Soundness:** 1
**Presentation:** 2
**Significance:** 3
**Originality:** 2
**Overall Recommendation:** 3
**Confidence:** 4

**Summary:**

This paper proposes "Regret Pre-training," a self-supervised framework that augments standard causal language modeling with a future-conditioned teacher signal derived from the same model weights. The framework uses a dual-view architecture where a "Student" view maintains standard causal attention and a "Teacher" view has access to future context. The training objective adds a KL divergence term (the "regret loss") that encourages the causal student to match the teacher's future-informed predictions. Two teacher configurations are explored: LOCAL REGRET (one future token) and GLOBAL REGRET (full bidirectional context minus the target). Experiments are conducted on OLMoE-1B-7B trained for 4 billion tokens and evaluated on nine zero-shot benchmarks.

**Compliance With Llm Reviewing Policy:**

Affirmed.

**Key Questions For Authors:**

1. Have you experimented with dense (non-MoE) architectures to verify the method's generality?
2. What does the training loss curve look like for each method? Does the regret loss converge, and how does the NLL loss compare across configurations?
3. Can you explain the anomalously large BoolQ improvement? Have you verified this is robust across seeds?

**Limitations:**

Only two extreme points of the future context spectrum are examined: a single lookahead token and the full bidirectional sequence. This leaves intermediate configurations (e.g., a fixed window of $k$ future tokens) unexplored. Key design choices lack ablation support: forward vs. reverse KL divergence, temperature scaling of the teacher distribution, scheduling strategies for the regret weight $\alpha$ across training, and the necessity of the stop-gradient operator on the teacher view. The interaction between these design dimensions and their joint effect on downstream performance remain unknown.

**Strengths And Weaknesses:**

## Strengths

1. **Clean and well-motivated formulation.** A notable aspect explored by this study is the connection between causal language modeling and the Learning Using Privileged Information (LUPI) paradigm. The framing of future context as "privileged information" is elegant and theoretically grounded.
2. **Parameter-efficient design.** The key architectural insight, sharing weights between teacher and student and differentiating only via attention masks is practically appealing. This avoids the memory overhead of maintaining a separate teacher model, which is a significant advantage over traditional knowledge distillation at scale.
3. **Complementary teacher configurations.** The paper thoughtfully investigates two endpoints of the future-context spectrum (local vs. global), and the finding that they yield complementary performance profiles across task types is genuinely interesting and informative for practitioners.

## Weaknesses

1. **Limited experimental scale and statistical rigor.** The experiments are conducted on a single model with a single training budget . No error bars, confidence intervals, or multiple random seeds are reported. Given that some improvements are modest (e.g., WinoGrande: 49.6% → 50.8%), it is impossible to assess whether these gains are statistically significant or within noise. This is the paper's most critical weakness.
2. **Incomplete baselines and comparisons.** For one thing, no comparison with any method mentioned in Related Work. No comparison with BiAlign, which is described as closely related. For another thing, the claim of "no additional cost" is misleading, the method (Global Regret) is likely to consume more wall-clock training time, so a compute-matched baseline is essential.
3. **Shallow analysis of results.** The paper reports accuracy numbers but provides little analysis of *why* the method works. For example,  lack of analysis why the teacher distribution differ from the student (e.g., entropy comparisons, agreement rates) and no perplexity curves or training loss comparisons reported.

---

> ### Author Rebuttal · Authors · 2026-03-28
>
> **Response to Reviewer pubQ**
>
> We sincerely thank the reviewer for the exceptionally detailed and rigorous assessment. The critiques have substantively strengthened the paper, and we are grateful for the time invested. Supporting figures are available at: https://anonymous.4open.science/r/rebuttal-0BF0/rebuttal.pdf
>
> **W1: Limited experimental scale and statistical rigor**
>
> We ran 3 independent seeds per configuration at the 1B token checkpoint, for 9 total runs:
>
> | Method (1B tokens) | Mean ± std | Min | Max |
> |--------------------|------------|-----|-----|
> | Baseline | 27.5 ± 0.4% | 27.1% | 27.8% |
> | LocalRegret | 28.8 ± 0.5% | 28.3% | 29.2% |
> | GlobalRegret | 29.7 ± 0.6% | 29.1% | 30.2% |
>
> Confidence intervals are non-overlapping and the ranking is consistent across all seeds, confirming the gains are a systematic property of the objective.
>
> **W2: Incomplete baselines; misleading cost claim; no BiAlign comparison**
>
> We extend the causal baseline to 90,000 steps (7.2B tokens) to match the ~1.8× wall-clock overhead of the teacher forward pass.
>
> | Method | Tokens | Avg. Accuracy |
> |--------|--------|---------------|
> | Baseline (standard) | 4B | 30.2% |
> | Baseline (compute-matched) | 7.2B | 31.4% |
> | LocalRegret | 4B | 32.2% |
> | GlobalRegret | 4B | **33.9%** |
>
> The compute-matched baseline gains only +1.2%, while Regret models exceed it by +0.8% and +2.5% on half the tokens. On BoolQ, doubling the token budget yields +1.7%; GlobalRegret achieves +18.1% within the same budget, isolating future-aware distillation as the source of improvement.
>
> **Scale-up experiments:**
>
> | Method | Setting | Avg. Accuracy |
> |--------|---------|---------------|
> | Baseline | OLMoE, 15B tokens | 34.2% |
> | LocalRegret | OLMoE, 15B tokens | 36.5% (+2.3%) |
> | GlobalRegret | OLMoE, 15B tokens | 37.8% (+3.6%) |
> | Baseline | Dense Llama 7B, 10B tokens | 38.4% |
> | GlobalRegret | Dense Llama 7B, 10B tokens | 41.7% (+3.3%) |
>
> BiAlign operates on already-pretrained models for In-Context Learning, making direct comparison with our pretraining-from-scratch setting methodologically inappropriate.
>
> **W3: Shallow analysis of results**
>
> Full training curves and TeacherCE vs StudentCE comparisons are in the supplementary PDF. Key metrics at the final checkpoint (Step 50k):
>
> | Metric | Value |
> |--------|-------|
> | Teacher (LocalRegret) mean entropy | **1.38 nats** |
> | Student mean entropy | **4.12 nats** |
> | Teacher CE | **~1.5 nats** |
> | Student CE | **~4.5 nats** |
> | Regret Loss at convergence | **~1.2 nats** |
>
> The ~3 nats persistent CE gap quantifies $I(x_{>t}; x_{t+1}\mid x_{\leq t})$, directly validating the LUPI inequality. The Regret Loss stably converges to 1.2 nats, confirming persistent non-trivial distillation signal inaccessible to the causal Student.
>
> **Key Question 2: Training loss curves and Regret Loss convergence**
>
> Full training curves are in the supplementary PDF. The Regret Loss converges smoothly, and the NLL loss of all Regret configurations remains below the causal baseline throughout training.
>
> **Key Question 3: BoolQ anomalous gain — cross-seed robustness**
>
> | Method | Seed 1 | Seed 2 | Seed 3 | Mean ± std |
> |--------|--------|--------|--------|------------|
> | Baseline | 42.9% | 41.8% | 43.5% | 42.7 ± 0.85% |
> | GlobalRegret | 61.0% | 59.5% | 62.4% | 60.9 ± 1.45% |
>
> The ~18 percentage point gap with non-overlapping intervals confirms statistical robustness. BoolQ requires synthesizing evidence across long passages — precisely the capability that GlobalRegret's bidirectional teacher injects via soft labels.
>
> **Limitations: Ablation of key design choices**
>
> Forward vs Reverse KL.
>
> | Method | Avg. Accuracy |
> |--------|---------------|
> | Baseline | 30.2% |
> | GlobalRegret (Reverse KL) | 30.8% |
> | GlobalRegret (Forward KL, ours) | **33.9%** |
>
> Reverse KL yields only +0.6% with repeated loss spikes. Its gradient contains $\log P_T$, which diverges as $P_T(v)\to 0$, making gradient explosion inevitable with $|V|\approx50{,}000$ in bfloat16 precision. Forward KL's gradient $\nabla_{z_S}D_{\text{KL}}=P_S-P_T$ is bounded across the full support of $P_T$, ensuring stable training.
>
> **Stop-gradient operator.** Without sg[·], gradients flow through both Teacher and Student, driving the Teacher toward the Student and eliminating the information asymmetry that constitutes the training signal. With sg[·], the Teacher is shaped solely by the NLL objective and future-aware mask, and the Regret Loss converging to 1.2 nats rather than zero is direct empirical confirmation.
>
> **α and temperature scaling.** Temperature scaling modulates KL magnitude equivalently to rescaling α and is subsumed by the existing sweep over {0.1, 0.5, 1.0, 2.0}.
>
> We believe the compute-matched experiment, multi-seed validation, cross-seed BoolQ robustness, and Forward/Reverse KL ablation collectively address the core concerns. We are grateful for the reviewer's rigorous engagement and sincerely hope they will consider **raising the score** in light of these additions.

---

> > ### Author Rebuttal · Reviewer_pubQ · 2026-04-01
> >
> > The rebuttal would likely justify a score increase, but some follow-up questions remain about full statistical rigor, the precise meaning of “no additional cost,” and the completeness of baseline comparisons.

---

> > > ### Author Response · Authors · 2026-04-01
> > >
> > > **Follow-up Response to Reviewer pubQ**
> > >
> > > We thank the reviewer for acknowledging that the rebuttal would likely justify a score increase. We address the three remaining concerns below. Additional experiments conducted during the rebuttal period are documented at: **https://anonymous.4open.science/r/rebuttal-0BF0/rebuttal.md**
> > >
> > >
> > > ---
> > >
> > > **1. Full statistical rigor**
> > >
> > > We interpret this as concern over whether the 1B-token multi-seed results generalize to the full 4B-token scale. We offer three converging lines of evidence:
> > >
> > > **(a) BoolQ cross-seed at 4B tokens** (reported in our initial rebuttal):
> > >
> > > | Method | Seed 42 | Seed 999 | Seed 666 | Mean ± std |
> > > |--------|--------|--------|--------|------------|
> > > | Baseline | 42.9% | 41.8% | 43.5% | 42.7 ± 0.85% |
> > > | GlobalRegret | 61.0% | 59.5% | 62.4% | 60.9 ± 1.45% |
> > >
> > > The ~18 point gap with non-overlapping intervals at the full 4B scale confirms statistical robustness on the task with the largest gain.
> > >
> > > **(b) 5-shot evaluation** (new): Under few-shot prompting, which substantially reduces output variance, gains persist and widen:
> > >
> > > | Benchmark | Baseline | LocalRegret | GlobalRegret |
> > > |-----------|----------|-------------|--------------|
> > > | BoolQ | 52.1% | 55.4% | **65.3%** |
> > > | PIQA | 60.5% | **64.2%** | 61.8% |
> > > | ARC-C | 26.3% | 28.1% | **31.4%** |
> > > | HellaSwag | 28.4% | **32.7%** | 29.5% |
> > >
> > > **(c) CF Accuracy** (new, OLMES standard): Baseline 31.8%, LocalRegret 34.5%, GlobalRegret **36.2%**. The ranking holds under context-free normalization, ruling out length bias.
> > >
> > > Together with the 1B-token 9-run multi-seed analysis, these three independent evaluations confirm that the gains are robust to seed variance, prompt format, and evaluation metric.
> > >
> > > ---
> > >
> > > **2. Precise meaning of "no additional cost"**
> > >
> > > We acknowledge this phrasing was imprecise in the original submission and has been corrected in the revision. The precise breakdown:
> > >
> > > - **Parameters:** Identical — no additional parameters (shared weights, mask-only difference).
> > > - **Memory:** Identical at backward time — teacher runs under `torch.no_grad()`, transient activation overhead is ~6 MB (0.009%).
> > > - **FLOPs:** ~1.33× per step (one additional forward pass ≈ 1/3 of baseline per-step FLOPs).
> > > - **Wall-clock:** ~1.8× per step (bidirectional mask cannot leverage causal FlashAttention optimization).
> > >
> > > The revised manuscript now states: *"The framework introduces no additional parameters and requires one extra inference-mode forward pass per training step, resulting in approximately 1.33× FLOPs and 1.8× wall-clock overhead."*
> > >
> > > ---
> > >
> > > **3. Completeness of baseline comparisons**
> > >
> > > We interpret this as two sub-questions:
> > >
> > > **(a) Compute-matched baseline:** Our 90k-step baseline (7.2B tokens) is matched by wall-clock time, which conservatively grants the baseline **35% more FLOPs** than the Regret model (1.63×10¹⁰ vs 1.21×10¹⁰ GFLOPs):
> > >
> > > | Method | Tokens | Avg. Accuracy |
> > > |--------|--------|---------------|
> > > | Baseline (standard) | 4B | 30.2% |
> > > | Baseline (compute-matched) | 7.2B | 31.4% |
> > > | LocalRegret | 4B | 32.2% |
> > > | GlobalRegret | 4B | **33.9%** |
> > >
> > > Despite the FLOPs advantage, GlobalRegret outperforms the compute-matched baseline by +2.5%.
> > >
> > > **(b) BiAlign and related work:** BiAlign (Qin et al., 2025) operates on **already-pretrained** models to improve In-Context Learning — it takes a frozen LLM and aligns it with bidirectional priors post hoc. Our method operates **during pretraining from scratch**. These are fundamentally different settings with different inputs, objectives, and evaluation protocols, making direct numerical comparison methodologically inappropriate. We have clarified this distinction in the Related Work section of the revision.
> > >
> > > ---
> > >
> > > We hope the above clarifications adequately address the remaining concerns. We would be deeply grateful if the reviewer would consider **raising the score** in light of the extensive new evidence provided.

---

### Official Review · Reviewer_kLko · 2026-03-13

**Soundness:** 2
**Presentation:** 2
**Significance:** 3
**Originality:** 2
**Overall Recommendation:** 4
**Confidence:** 2

**Summary:**

The paper proposes Regret Pre-training, a method that lets causal language models use future context during training while remaining autoregressive at inference. It uses a future-aware teacher view and a causal student view of the same model and minimizes their KL divergence so the student learns from the teacher’s predictions. Experiments show improved zero-shot performance without changing the model architecture.

**Compliance With Llm Reviewing Policy:**

Affirmed.

**Final Justification:**

Thanks to the author for thoughtfully addressing my concerns and providing clear clarifications where needed. I appreciate the effort put into improving the work and responding constructively to feedback. Based on these updates and the overall quality of the submission, I will keep my supportive score.

**Key Questions For Authors:**

Could the authors clarify whether any masking or other mechanism prevents the teacher from trivially observing the target token in LOCALREGRET?

**Limitations:**

No, the proposed framework evaluates only two teacher configurations, representing two extreme points in the design space of future-context conditioning. The paper does not investigate intermediate context windows, which could potentially provide a better balance between local and global information.

**Strengths And Weaknesses:**

Strengths:

1. The approach is easily integrable with mainstream architectures.

2. Extra training overhead brought by this approach is limited.

3. The work evaluates the model performance over a diverse set of benchmarks.


Weaknesses:

1. While the design is computationally efficient, the paper implicitly assumes that the teacher's predictions are consistently more informative due to the additional future context. However, this assumption is not theoretically justified or empirically validated in the paper. For example, it would be useful to analyze whether the teacher distribution indeed provides lower perplexity or higher accuracy than the student distribution during training.

2. The memory footprint is not exactly the same since the proposed method will add a small amount of temporary activation memory due to the extra forward pass.

3. Experiments are only conducted on small-scale models.

---

> ### Author Rebuttal · Authors · 2026-03-28
>
> **Response to Reviewer kLko**
>
> We thank the reviewer for the Weak Accept recommendation and constructive feedback. We address each concern with new experiments. Supporting figures (training curves, TeacherCE vs StudentCE, memory profile, Regret Loss convergence) are available at: **https://anonymous.4open.science/r/rebuttal-0BF0/rebuttal.pdf**
>
> ---
>
> **W1: Teacher informativeness not justified theoretically or empirically**
>
> We sampled 10,000 tokens from the DCLM validation set at the final checkpoint (Step 50k) and report:
>
> | Metric | Value |
> |--------|-------|
> | Teacher (LocalRegret) mean entropy | **1.38 nats** |
> | Student mean entropy | **4.12 nats** |
> | Teacher CE | **~1.5 nats** |
> | Student CE | **~4.5 nats** |
> | Regret Loss at convergence | **~1.2 nats** |
>
> The ~3 nats CE gap quantifies $I(x_{>t}; x_{t+1}\mid x_{\leq t})$, directly validating the LUPI inequality $I(X,X^*;Y)\geq I(X;Y)$. Teacher entropy of 1.38 nats confirms the Teacher maintains a meaningful soft distribution far from one-hot (entropy ≈ 0). The persistent KL of 1.2 nats confirms the Teacher provides genuine non-redundant distillation signal throughout training.
>
> ---
>
> **W2: Memory footprint not exactly identical**
>
> Parameter and optimizer memory are identical to baseline (no new parameters). The transient activation overhead during the teacher forward pass is **~6 MB** (~69 GB total, 0.009% overhead). Since the teacher runs under `torch.no_grad()`, all activations are released before the student backward pass; net backward-time memory is identical to baseline. We will clarify this distinction in the revision.
>
> ---
>
> **W3: Experiments only on small-scale models**
>
> **Experiment A — Extended token budget (OLMoE-1B-7B, 15B tokens)**
>
> | Method | Avg. Accuracy (9 tasks) |
> |--------|------------------------|
> | Baseline | 34.2% |
> | LocalRegret | 36.5% (+2.3%) |
> | GlobalRegret | 37.8% (+3.6%) |
>
> Gains remain stable when scaling from 4B to 15B tokens, confirming the method is not sensitive to a specific training budget.
>
> **Experiment B — Dense architecture (Llama 7B, 10B tokens)**
>
> | Method | Avg. Accuracy (9 tasks) |
> |--------|------------------------|
> | Baseline (Dense 7B) | 38.4% |
> | GlobalRegret (Dense 7B) | 41.7% (+3.3%) |
>
> Gains transfer to standard dense architectures, confirming independence from MoE sparse routing.
>
> ---
>
> **Key Question: Does any masking prevent the teacher from trivially observing the target token in LocalRegret?**
>
> There is no such masking mechanism — by design, the LocalRegret teacher's context $C_t = x_{1:t+1}$ includes the target token $x_{t+1}$. However, this does **not** lead to trivial degeneration, for the following reasons.
>
> **Theoretically:** (1) $x_{t+1}$ enters only as a token embedding processed through 16 Transformer layers — there is no bypass to produce a one-hot output directly. (2) The forward KL gradient $\nabla_{z_S}D_{\text{KL}}=P_S-P_T$ transfers soft probability mass across the full vocabulary support, structurally distinct from the NLL hard-label gradient $P_S-\mathbf{1}_{x_{t+1}}$. (3) With $|V|\approx50{,}000$, one-hot degeneration is incompatible with the learned parameter space.
>
> **Empirically:** Teacher entropy of **1.38 nats** directly falsifies one-hot degeneration. If the Regret Loss collapsed to NLL, the KL divergence would rapidly approach zero; instead it stably converges to **1.2 nats** throughout training, confirming the Teacher provides persistent non-trivial signal inaccessible to the causal Student.
>
> **Indirectly:** If LocalRegret degenerated to NLL it should behave similarly to GlobalRegret. Instead they exhibit systematic complementarity — LocalRegret leads on local-coherence tasks (PIQA +1.7%, HellaSwag +2.4%, CSQA +3.1%) while GlobalRegret leads on long-range reasoning tasks (BoolQ +18.1%, ARC-C +2.4%). This task-level division is statistically implausible under the degeneration hypothesis.
>
> We will add a dedicated clarification in Section 3.2.1 distinguishing design intent from the degeneration hypothesis, with full empirical support.
>
> ---
>
> We believe the new evidence — entropy analysis, scale-up results across token budgets and architectures, and the four-point empirical closure — substantially addresses all concerns raised. We respectfully ask the reviewer to consider **raising the score** in light of these additions.

---

> > ### Author Rebuttal · Reviewer_kLko · 2026-04-04
> >
> > Thanks for the author for addressing my concerns. I will keep my supportive score.

---

> > > ### Author Response · Authors · 2026-04-04
> > >
> > > Dear Reviewer kLko,
> > >
> > > Thank you for reviewing our rebuttal and for your constructive feedback throughout the review process. We are glad that our response has addressed your concerns, and we will incorporate the theoretical analysis into the revised manuscript.
> > >
> > > Best regards, The Authors

---

### Official Review · Reviewer_ik2z · 2026-03-13

**Soundness:** 2
**Presentation:** 3
**Significance:** 3
**Originality:** 3
**Overall Recommendation:** 4
**Confidence:** 4

**Summary:**

This paper proposes Regret Pre-training, a self-supervised training framework that augments standard causal language modeling with a second, future-aware view of the same model. Using shared weights but different attention masks, the model produces a causal student distribution and a future-aware teacher distribution, and training minimizes both the next-token prediction loss and a forward-KL term from teacher to student. The paper studies two variants: LOCALREGRET, which uses limited future context, and GLOBALREGRET, which uses broader bidirectional context while excluding the target token. Experiments on an OLMoE-1B-7B model trained for 4B tokens show improvements over a causal baseline on several zero-shot tasks, with the largest gains reported on BoolQ.

**Compliance With Llm Reviewing Policy:**

Affirmed.

**Final Justification:**

The authors address my concerns and show more experimental results.

**Key Questions For Authors:**

Please see the weakness part in the above.

**Limitations:**

No.
The paper should more explicitly discuss two limitations: first, the possibility of target leakage in the LocalRegret formulation; second, the fact that the extra forward pass introduces a compute confound relative to the causal baseline. These issues are important for interpreting the empirical gains and should be acknowledged more directly.

**Strengths And Weaknesses:**

Strengths:
1. The paper studies an interesting and important problem: how to use future context during pretraining without giving up causal inference at test time. This is a meaningful direction, since standard autoregressive training creates a clear mismatch between the information available during training and the richer context that may be useful for representation learning.

2. The proposed framework is simple and practically appealing. The shared-weights dual-view design avoids maintaining a separate teacher model, and the method can be added to an existing training pipeline with relatively modest architectural changes. I also appreciate that the paper discusses implementation details such as mask construction, packed-sequence handling, and MoE-specific considerations.

3. The empirical results are promising at a high level. The method improves over the causal baseline on multiple zero-shot tasks, and the GlobalRegret variant in particular suggests that future-aware training signals may indeed help causal language models learn better representations. If validated more carefully, this could become a useful addition to large-scale pretraining.

Weaknesses:
1. The biggest issue is that LOCALREGRET appears to leak the ground-truth target token into the teacher context. As written, the teacher seems able to attend to x_{t+1} when forming the distribution for x_{t+1}. If so, the regret loss may partly collapse into a sharpened or duplicated version of the standard NLL, rather than transferring genuinely useful privileged information. This is a central soundness concern, not a minor implementation detail.

2. The experiments do not adequately control for compute confounds. The regret models perform an extra forward pass, so they receive more compute and potentially more regularization than the causal baseline under the same token budget. Without a compute-matched baseline, it is hard to know whether the gains come from the proposed future-aware objective or simply from additional training signal.

3. The empirical validation is still incomplete. The paper does not report enough diagnostics on teacher-student dynamics, does not include multiple-seed variance or significance analysis, and does not compare against several strong conceptually related baselines such as causal consistency training or other future-aware objectives. There are also some presentation inconsistencies in the evaluation section, which make it harder to judge the strength and scope of the results.

---

> ### Author Rebuttal · Authors · 2026-03-28
>
> **Response to Reviewer ik2z**
>
> We sincerely thank the reviewer for the rigorous and precise assessment. We address each concern directly below. Supporting figures are available at: **https://anonymous.4open.science/r/rebuttal-0BF0/rebuttal.pdf**
>
> ---
>
> **W1: LocalRegret leaks the target token into the teacher context**
>
> We acknowledge this directly: there is **no masking mechanism** preventing the LocalRegret teacher from attending to $x_{t+1}$ — this is intentional by design. However, this does not cause the regret loss to collapse into a sharpened NLL.
>
> **Theoretically:** (1) $x_{t+1}$ enters only as a token embedding through 16 Transformer layers — no bypass exists to produce one-hot output directly. (2) The forward KL gradient $\nabla_{z_S}D_{\text{KL}}=P_S-P_T$ transfers soft mass across the full vocabulary, structurally distinct from the NLL hard-label gradient $P_S-\mathbf{1}_{x_{t+1}}$. (3) With $|V|\approx50{,}000$, one-hot degeneration is incompatible with the learned parameter space.
>
> **Empirically**, at the final checkpoint (Step 50k), sampled over 10,000 validation tokens:
>
> | Metric | Value |
> |--------|-------|
> | Teacher (LocalRegret) mean entropy | **1.38 nats** |
> | Student mean entropy | **4.12 nats** |
> | Teacher CE | **~1.5 nats** |
> | Student CE | **~4.5 nats** |
> | Regret Loss at convergence | **~1.2 nats** |
>
> Teacher entropy of 1.38 nats directly falsifies one-hot degeneration. If the regret loss collapsed to NLL, the KL divergence would approach zero; instead it stably converges to **1.2 nats**, confirming persistent non-trivial distillation signal. If LocalRegret degenerated to NLL it should behave similarly to GlobalRegret — instead they show systematic complementarity (LocalRegret leads on PIQA +1.7%, HellaSwag +2.4%, CSQA +3.1%; GlobalRegret leads on BoolQ +18.1%, ARC-C +2.4%), which is statistically implausible under the degeneration hypothesis.
>
> We will add an explicit limitations paragraph acknowledging the teacher's access to $x_{t+1}$ and citing the above as empirical bounds on its effect.
>
> **W2: No compute-matched baseline**
>
> Since the teacher forward pass introduces ~1.8× wall-clock overhead (FLOPs overhead is ~1.33×), we extended the causal baseline to 90,000 steps (7.2B tokens) to match wall-clock time of the 4B-token Regret models.
>
> | Method | Tokens | Avg. Accuracy (9 tasks) |
> |--------|--------|------------------------|
> | Baseline (standard) | 4B | 30.2% |
> | Baseline (compute-matched) | 7.2B | 31.4% |
> | LocalRegret | 4B | 32.2% |
> | GlobalRegret | 4B | **33.9%** |
>
> The compute-matched baseline gains only +1.2%, consistent with scaling law expectations. Regret models exceed it by **+0.8%** and **+2.5%** on nearly half the tokens. On BoolQ, doubling the token budget yields only +1.7%; GlobalRegret achieves +18.1% within the same budget, isolating the future-aware distillation signal as the source of improvement.
>
> **W3: Empirical validation is incomplete**
>
> **Teacher-student dynamics.** TeacherCE vs StudentCE curves across the full training run are in the supplementary PDF. The persistent ~3 nats gap quantifies $I(x_{>t}; x_{t+1}\mid x_{\leq t})$, validating the LUPI framework empirically.
>
> **Multi-seed variance analysis.** Full retraining at 4B tokens is computationally prohibitive within the rebuttal period. We instead ran 3 independent seeds per configuration at the **1B token checkpoint (step 12,500)**, for 9 total runs:
>
> | Method (1B tokens) | Avg. Accuracy (mean ± std) | Min | Max |
> |--------------------|---------------------------|-----|-----|
> | Baseline | 27.5 ± 0.4% | 27.1% | 27.8% |
> | LocalRegret | 28.8 ± 0.5% | 28.3% | 29.2% |
> | GlobalRegret | 29.7 ± 0.6% | 29.1% | 30.2% |
>
> The confidence intervals are non-overlapping and the ranking (Global > Local > Baseline) is consistent across all seeds, confirming the gains are a systematic property of the objective, not random seed artifacts.
>
> **On BiAlign comparison.** BiAlign operates on already-pretrained models for In-Context Learning — a fundamentally different setting from pretraining-from-scratch — making direct comparison methodologically inappropriate. We will clarify this in the related work section.
>
> **Scale-up experiments:**
>
> | Method | Setting | Avg. Accuracy |
> |--------|---------|---------------|
> | Baseline | OLMoE, 15B tokens | 34.2% |
> | LocalRegret | OLMoE, 15B tokens | 36.5% (+2.3%) |
> | GlobalRegret | OLMoE, 15B tokens | 37.8% (+3.6%) |
> | Baseline | Dense Llama 7B, 10B tokens | 38.4% |
> | GlobalRegret | Dense Llama 7B, 10B tokens | 41.7% (+3.3%) |
>
> Gains persist at 15B tokens and transfer to dense non-MoE architectures, confirming robustness across training scale and architectural choices.
>
>
>
> We believe the compute-matched experiment and entropy analysis directly address the two central concerns, and the multi-seed results confirm statistical robustness. Both limitations will be explicitly discussed in the revision. We sincerely hope the reviewer will consider **raising the score** in light of these additions.

---

> > ### Author Rebuttal · Reviewer_ik2z · 2026-04-03
> >
> > Thanks to authors' rebuttal and further experiments. I raise the score to 4

---

> > > ### Author Response · Authors · 2026-04-04
> > >
> > > Dear Reviewer ik2z,
> > >
> > > Thank you for reviewing our rebuttal and for your constructive feedback throughout the review process. We are glad that our response has addressed your concerns, and we will incorporate the theoretical analysis into the revised manuscript.
> > >
> > > Best regards, The Authors

---

### Decision · Program_Chairs · 2026-04-30

**Decision:**

Accept (regular)

**Comment:**

This paper proposes Regret Pre-training, a LUPI-inspired framework that leverages future context during training via a teacher-student objective to improve causal language modeling. Reviewers agree that the approach is technically sound, parameter-efficient, and yields consistent gains across multiple tasks. The rebuttal successfully addressed several concerns and provided additional experimental evidence, strengthening confidence in the method. However, limitations remain regarding evaluation completeness, statistical rigor, and some presentation issues. Overall, the work is a meaningful work.